# Research Progress on the Quality, Extraction Technology, Food Application, and Physiological Function of Rice Bran Oil

**DOI:** 10.3390/foods13203262

**Published:** 2024-10-14

**Authors:** Wengong Huang, Baohai Liu, Dongmei Shi, Aihua Cheng, Guofeng Chen, Feng Liu, Jiannan Dong, Jing Lan, Bin Hong, Shan Zhang, Chuanying Ren

**Affiliations:** 1Safety and Quality Institute of Agricultural Products, Heilongjiang Academy of Agricultural Sciences, Harbin 150086, China; huangwengong1736@163.com (W.H.); shslbh@163.com (B.L.); shidongmei@126.com (D.S.); aihcheng@163.com (A.C.); hljjiance@163.com (G.C.); 13766881790@126.com (F.L.); jiannan_dong@163.com (J.D.); lanjing1968@163.com (J.L.); 2Key Laboratory of Quality and Safety of Cereals and Their Products, State Administration for Market Regulation, Harbin 150086, China; 3Food Processing Research Institute, Heilongjiang Academy of Agricultural Sciences, Harbin 150086, China; gru.hb@163.com (B.H.); zhangshanfood@163.com (S.Z.)

**Keywords:** rice bran oil, nutrition, new oil extraction, improve food characteristics, biological activity

## Abstract

Rice bran oil is recommended by the World Health Organization as one of the three major healthy edible oils (along with corn and sesame oils), owing to its unique fatty acid composition and functional components. This study screened, organized, and analyzed a large number of studies retrieved through keyword searches, and investigated the nutritional value and safety of rice bran oil. It reviews the stability of raw rice bran materials and the extraction and refining process of rice bran oil and discusses food applications and sub-health regulations. Research has found that a delayed stabilization treatment of rice bran seriously affects the overall quality of rice bran oil. Compared with traditional solvent extraction, the new extraction technologies have improved the yield and nutritional value of rice bran oil, but most of them are still in the research stage. Owing to the lack of economical and applicable supporting production equipment, extraction is difficult to industrialize, which is a challenging research area for the future. Rice bran oil has stronger antioxidant stability than other edible oils and is more beneficial to human health; however, its application scope and consumption are limited owing to the product price and lack of understanding. Rice bran oil has significant antioxidant, anti-inflammatory, anti-cancer, hypoglycemic, lipid-lowering, and neuroprotective effects. Further exploratory research on other unknown functions is required to lay a scientific basis for the application and development of rice bran oil.

## 1. Introduction

Rice is the world’s second-largest food crop after wheat. The crop is grown in 114 countries, and more than 650 million tons are produced annually. Although China’s rice planting area ranks second globally, its total output ranks first, and its yield per unit area is approximately 50% higher than the world average [1]. By 2023, China’s total rice production is expected to reach 210 million tons, accounting for approximately one-third of the world’s output. Many types of rice are cultivated in China, and the proportion of irrigated areas is much higher than the world average [2]. These varieties are famous for their high yields and high quality. Rice is the world’s leading cereal crop and serves as the staple food for approximately half of the global population [3].

Rice bran is an underutilized byproduct of brown rice milling, accounting for 5% to 10% of the total rice mass fraction. The crop is enriched in lipids (12–23%), protein (14–16%), and crude fiber (8–10%) [4]. It also contains abundant non-starch polysaccharides, vitamins, minerals, phenolic acids, flavonoids, tocopherols, tocotrienols, and γ-glutamine [5]. One of the most common utilizations of rice bran is the extraction of rice bran oil. Among all vegetable oils, rice bran oil is rich in antioxidant compounds that provide several beneficial properties [6]. The lipids are extracted into the pale yellow and translucent rice bran oil, which has a slightly nutty taste. Rice bran oil is typically composed predominantly of triglycerides (81–84%), with small quantities of monoacylglycerol (1–2%), diacylglycerol (2–3%), free fatty acids (FFAs, 2–6%), waxes (3–4%), glycolipids (0.8%), phospholipids (1–2%), and unsaponifiable matter (4%) [7]. The World Health Organization (WHO), China Grain and Oil Association, American Heart Association, Indian Council of Medical Research, and National Institute of Nutrition of India have all described rice bran oil as a “healthy” edible oil [8].

Owing to the lack of systematic and comprehensive discussions on the characteristics and applications of rice bran oil in the literature, in this study, the nutritional and safety qualities of rice bran oil were analyzed, and the stabilization of rice bran raw materials and the extraction and refining process of rice bran oil were reviewed. Finally, its application in food and sub-health regulations is discussed.

## 2. Methodology

In this review, the nutritional characteristics, raw material safety, extraction and refining technology, application in food, and physiological function of rice bran oil were comprehensively and systematically studied and discussed. Key databases, such as Web of Science, PubMed, X-MOL, MDPI, and Springer, were selected to ensure a comprehensive and in-depth search of relevant studies. The keywords of the literature search were rice bran oil, nutrition, stabilized rice bran, extraction, application, and biological activity. First, duplicate and non-English studies were mostly removed, and the title and abstract were carefully checked to confirm whether they met the criteria. The studies were then classified according to the predetermined titles, and studies from 2020 to 2024 were selected. Finally, the selected studies were summarized and analyzed. A total of 98 out of 268 articles were selected, forming a solid basis for this review.

## 3. Nutritional Quality of Rice Bran Oil

Rice bran oil is a nutrient-rich vegetable oil. The fatty acid composition, γ-glutamate, vitamin E, plant sterols, and phenolic compounds are conducive to human absorption, with a rate exceeding 90%. The primary nutritional components are listed in Table 1. 

The contents of unsaturated fatty acids in rice bran oil extracted from different varieties or strains of rice vary. In one study, the three most abundant and health-related unsaturated fatty acids (*oleic acid*, *linoleic acid*, and *alpha-linolenic acid*) were freely present in rice bran oil from the nine parent lines (two sterile male lines and seven male lines) and seven hybrid rice lines. The hybrid lines had the highest oil contents, whereas the male lines had the highest contents of two of the three free unsaturated fatty acids (*linoleic* and *oleic acids*) [12]. After fermentation of rice bran with lactic acid bacteria (*Lactobacillus acidophilus*, *L. bulgaricus*, and *Bifidobacterium bifidum*), the nutritional content of rice bran oil changed significantly, with increases in total protein (29.52%), fat (5.38%), ash (48.47%), crude fiber (38.96%), and water (61.04%). The carbohydrate content was reduced (36.61%). Increased contents of free amino acids, total phenols, tannins, flavonoids, γ-oryzanol, vitamin E, and antioxidant activities have been described [13].

### 3.1. Free Fatty Acids (FFA)

The unsaturated fatty acid contents of edible vegetable oils from different raw materials differ significantly (Figure 1). The saturated, monounsaturated, and polyunsaturated fatty acids ratio in rice bran oil is 0.6:1.1:1, which is close to the golden share ratio advocated by the WHO and is a hallmark of “healthy” edible oils [14]. The triglycerides in rice bran oil are mainly composed of the following three fatty acids: palmitic acid (C16:1), oleic acid (C18:1), and linoleic acid (C18:2). These fatty acids accounted for more than 90% of the total fatty acids in rice bran oil. Oleic acid is the most abundant FFA in rice bran oil, accounting for 42% of the total triglycerides, followed by linoleic acid (32%); however, myristic and stearic acids are relatively limited [9]. The fat content of Indian rice bran is as high as 24% and is mainly composed of oleic acid (48.48%), linoleic acid (35.26%), palmitic acid (14.54%), and FFAs (8.15%), with abundant antioxidants, including glutamic acid, tocopherol, and tocopherol [15]. 

### 3.2. γ-Oryzanol

In addition to fatty acids, rice bran oil contains non-saponifiable lipids with a high nutritional value. The most important of these lipids is γ-oryzanol. Oryzanol lipid-binding protein is composed of ferulic acid and phytosterols. The γ-oryzanol constituent 2,4-methylenecycloartanyl ferulate is abundant in purified rice bran oil. We have also described two additional novel γ-oryzanol compounds in rice bran oil, as follows: cyclobranyl ferulate and cyclosadyl ferulate [16]. Crude rice bran oil contains methyl ferulate (0.3%), cycloxybromelane ferulate (0.14%), 24-methylene cycloxybromelane ferulate (0.56%), ferulate-canolasterol (0.49%), and β-sitosterol ferulate (0.24%) [17]. The content of γ-oryzanol in rice bran oil ranges from 0.20% to 2.72%, owing to different rice varieties, weather, and planting areas [18]. γ-Oryzanol is unstable under high-temperature conditions (above 100 °C) and can also be lost under the influence of chemical solvents [19].

### 3.3. Vitamin E

Plant sterols comprise 4-methyl-free sterols, 4-methyl-sterols, and 4,4′-dimethyl-sterols. The main non-methyl-sterols are Β-sitosterol, stigmosterol, and rapeseed sterol [20]. Plant sterols, which mainly exist in plant seeds, are structurally similar to animal sterols. The only differences were the number of methyl groups associated with c-4 and the number of c-11 side chains [21]. Slight differences in these side chains result in differences in physiological functions [22]. Rice bran oil contains a large number of phytosterols, including β-sitosterol (0.90–1.70%), canola sterol (0.50–0.66%), and stigasterol (0.27–0.25%). These sterols have antioxidant properties and constitute a central portion of the non-juice component of nutritional supplements [23].

### 3.4. Phytosterol

Plant sterols are divided into 4-methyl-free sterols, 4-methyl-sterols, and 4,4′-dimethyl-sterols. The main non-methyl-sterols are Β-sitosterol, stigmosterol, rapeseed sterol, and so on [24]. The structures of plant sterols, which mainly exist in plant seeds, are similar to those of animal sterols. The only difference was the number of methyl groups associated with c-4 and the difference in the c-11 side chain. Slight differences in these side chains result in different physiological functions [25]. Rice bran oil contains a large number of phytosterols, including β-sitosterol (0.90–1.70%), canola sterol (0.50–0.66%), and stigasterol (0.27–0.25%). These sterols have antioxidant properties and constitute the main portion of the non-juice component of nutritional supplements [26].

### 3.5. Phenolic Compounds

Rice bran oil is rich in phytochemicals, including vitamins, cereals, fatty acids, and phenolic compounds [27]. Phenolic compounds in rice bran oil may reduce oxidative stress and cardiovascular disease risk factors by lowering low-density lipoprotein cholesterol levels [28]. In one study, the content of free phenols in bran oil increased gradually with increasing storage time, the contents of combined phenols and total phenols first increased and then decreased, and antioxidant activity was also significantly affected. The total phenolic content of the rice bran oil did not exceed 2 mg/100 g [29].

## 4. Safety Quality of Rice Bran Oil

Rice bran has good prospects for processing into rice bran oil because it is rich in unsaturated fatty acids (*oleic acid*, *linoleic acid*, etc.) and healthy bioactive substances. However, the hydrolysis and oxidation rancidity of these lipids during storage are significant problems that directly affect the quality of rice bran oil.

### 4.1. Oil Oxidation Mechanism

There are three mechanisms for oil oxidation (Figure 2) [30]. The first is autooxidation, which is the reaction of free radicals between activated alkene-containing substrates and ground-state oxygen (Figure 2a). Autooxidation includes the following three stages: chain initiation, chain transfer, and chain termination. The second mechanism is photosensitive oxidation, which is the direct oxidation reaction between unsaturated double bonds and singlet oxygen under the action of a photosensitizer (Figure 2b). The reaction forms a six-membered ring transition state, forming a trans-configuration of hydroperoxide. The third mechanism is reactions that occur in fats, as catalyzed by two types of enzymes (Figure 2c). One is the specific action of fat oxidase on polyunsaturated fatty acids with 1,4 cis and cis-pentadiene structures. The other is the keto-type spoilage caused by the action of enzymes produced by some microorganisms during reproduction that mainly occurs between the α- and β-carbon positions of saturated fatty acids and is also known as the β-oxidation reaction. Oleic acid and linoleic acid account for more than 70% of the FFAs in rice bran oil and are the main causes of oxidation of rice bran oil (Figure 3). In the oleic acid molecule, the hydrogens of the 8- and 11-position carbon atoms have the same activity; therefore, it can generate two different free radicals and four kinds of hydroperoxides. In the linoleic acid molecule, because only the 11-position hydrogen is particularly active, only one free radical and two hydroperoxides are generated. 

FFA content and peroxidation value (PV) are essential indices for evaluating the degree of rancidity of rice bran [31]. In the oil oxidation mechanism, the oxidation rate of rice bran oil is directly proportional to the number of double bonds in unsaturated fatty acids, oxygen partial pressure, temperature, surface area, water activity, light, lipase activity, and some metal elements. Therefore, the stabilization treatment of rice bran mainly reduces the hydrolytic rancidity reaction and inhibits the growth of FFAs by passivating lipase activity and reducing water activity (Table 2). These actions prolong the shelf life of rice bran [32]. The latter authors reported that after 14 days of storage at room temperature, the levels of bioactive compounds and functional lipids (25 acylglycerols and 53 phospholipids) were significantly reduced. These significant reductions indicated a rancid process. Phospholipid and glycerolipid metabolic pathways are key for lipid metabolism [33]. Therefore, stabilizing rice bran is key to improving the safety of rice bran oil.

### 4.2. Stabilization of Rice Bran

Rice bran is rich in nutrients and contains many bioactive compounds that benefit health. However, the rapid deterioration of rice bran limits its development and utilization. Hydrolytic rancidity is the main obstacle in rice bran extraction. The only way to prevent the hydrolytic rancidity of rice bran is to promote enzyme inactivation [34]. Microwave pretreatment stabilizes rice bran and increases its phytochemical content. In one study, when rice bran was treated with 440 W of microwave radiation for 2.5 min, increases were evident in antioxidant activity (0.5-fold), total phenolic contents (1.3-fold), total flavonoid contents (0.9-fold), total γ-oryzanols (1.6-fold), and total phytosterols (1.4-fold). Furthermore, phytochemicals were enhanced, especially trans-p-coumaric acid (10.3-fold) and kaempferol (8.6-fold) [35]. In the microwave-assisted stabilization of rice bran from the Ariete, Teti, and Luna varieties of rice (three commercial Carolino rice cultivars, Ariete, Luna, and Teti, grown in Portugal in the Mondego, Sado, and Tejo River valleys, respectively), oil was extracted, and the effects of rice bran stability on γ-aminobutyric acid (GABA) and γ-glutamate compounds were studied. The stability of LUNA rice bran did not significantly affect these two components. The content of γ-glutamate in the ARIETE and TETI varieties decreased by 34.4% and 24.2%, respectively, and GABA increased by 26.5% and 47.0%, respectively. However, the γ-glutamate content of rice bran oil is not affected by its stability of rice bran [36]. 

In another study, four methods of treating rice bran were compared, including steaming, hot air drying, microwave treatment, and a combination of microwave and heat treatment, and the rice bran was stored at room temperature for 50 d. The results showed that the combined microwave treatments had the best effects on the oxidation stability of rice bran; FFAs and PV were the lowest, as was the increment. Microwave-treated rice bran has the highest total phenol and γ-glutamate, antioxidant activity, and half-maximal inhibitory concentration (IC50) value [31]. The stabilization of rice bran using infrared radiation heating can improve the quality of rice bran oil. In one study, when rice bran was stabilized using infrared radiation at 125 and 135 V for 5 to 10 min, lipase activity was the lowest (inhibition rate 93~96%), γ-oryzanol and α-tocopherol contents were not significantly changed, and the increases in FFA content and PV were completely inhibited after storage at 38 °C for eight weeks [37].

## 5. Extraction and Refining of Rice Bran Oil

In industrial production, the traditional process for extracting rice bran oil is solvent extraction. The crude oil obtained is refined through processes such as deacidification, deodorization, decolorization, and degumming to obtain rice bran oil. The byproducts of the refining process can be used to extract various active substances, such as γ-oryzanol, tocopherol, and plant sterols (Figure 4).

### 5.1. Extraction Technology of Rice Bran Oil

The extraction of rice bran oil is usually achieved through solvent extraction. N-hexane is the primary solvent because of its high solubility, low boiling point, and low cost. However, residual organic solvents can easily cause poisoning in humans. It was reported that with isopropyl alcohol and cyclohexane as extraction solvents instead of n-hexane, the peroxide, acid, iodine, and fatty acid compositions of rice bran oil were significantly lower than those of n-hexane. In contrast, the glutamine and total sterol contents increased to 2.7% and 5.1%, respectively [38]. The liquid–solid extraction of rice bran oil with ethyl acetate and ethanol was more compatible with the oil compounds in the rice bran and had a lower viscosity. Ethyl acetate had a higher extraction rate at the beginning and reached its highest yield within a short time [39] (Table 3).

Mechanical pressing is a conventional extraction technique that does not involve heat treatment or any organic solvent and is less expensive than solvent extraction. In addition, the oil extracted using this method has better nutritional characteristics and is safer, thereby increasing the value of cold-pressed vegetable oil. New combined technologies have emerged to improve the efficiency and quality of rice bran oil. These include enzyme-assisted extraction, ultrasound-assisted water extraction, supercritical and subcritical CO_2_ extraction, microwave-assisted extraction, and subcritical fluid technology. All these have broad application prospects [40].

#### 5.1.1. Enzyme-Assisted Extraction

The addition of specific enzymes (cellulase, pectinase, and protease) to destroy the cell wall and release active substances significantly improves the extraction rate. Most of the properties are the same as those of the solvent extraction of rice bran oil, but the content of colored substances is lower and the acidity (FFA) is higher [41]. The highest yield of rice bran oil was 75% using alkaline protease, hemicellulase, pentosan complex enzyme, and papain [42]. Micron rice bran oil bodies were obtained by combining xylanase with the plant extraction enzyme at a ratio of 2:1 (*w*/*w*), an additional amount of 3%, a pH of 5.0, and an enzymolysis time of 1.5 h [43]. The grain size of rice bran oil extracted using enzyme-assisted NaHCO_3_ was smaller, and the content of the β-folding structure was higher, which was conducive to the stretching and aggregation of protein spatial structure and better physical stability [44]. The specificity of the enzymatic reaction was high, the reaction conditions were mild and easy to control, and the extraction time was shortened. The surface protein of the rice bran oil body is not destroyed, the stability of the rice bran oil is good, its structure is complete, and the availability of physiologically active components can be improved.

#### 5.1.2. Ultrasonic-Assisted Extraction

Ultrasonic waves transmit high-frequency signals and convert them into high-frequency mechanical oscillations. The intense flow of the solvent produces tiny bubbles, and the closure of the bubbles produces shock waves that improve the permeability of the material surface. The extraction rate of rice bran oil was close to that of Soxhlet extraction when the ultrasonic-assisted extraction technology was at pH 12, the temperature was 45 °C, the stirring speed was 800 r/min, the stirring time was 15 min, the ultrasonic treatment time was 70 min, and the ultrasonic treatment temperature was 25 °C [45]. However, the content of ω-6 polyunsaturated fatty acids (especially 9, 12-octadecadienoic acid (Z,Z)-fatty acids, namely, linoleic acid (64.19%)), in rice bran oil extracted through ultrasonic-assisted extraction was higher than that of hexane extraction [46]. Ultrasound-assisted solvent extraction shortened the homeostasis time from 15 min to 1 min and significantly increased the yield of rice bran oil and γ-oryzanol [47]. The technology has a low temperature, saves considerable energy, is a simple extraction process, improves the extraction rate, and reduces production costs. Low-temperature extraction does not change the chemical composition and structure of nutrients, promotes the extraction of active substances, and improves the nutritional value of rice bran oil.

#### 5.1.3. Supercritical and Subcritical CO_2_ Extraction

In the supercritical state, the supercritical fluid comes into contact with the substance to be separated, and the components of polarity, boiling point, and molecular weight are selectively and successively extracted. When the pressure exceeded the critical point, the CO_2_ fluid significantly increased the conversion rate of phytosterols to 93.2% and the retention rate of vitamin E to 80.0% [48]. The yield of the subcritical CO_2_ extraction of rice bran oil was lower than that of hexane extraction, but the tocopherol and glutarol contents in the rice bran oil extracted were approximately 10 times those of hexane extraction, and the FFA and peroxide values were significantly reduced [49]. The contents of total flavonoids and total polyphenols increased, reaching 61.28 μmol GAE/g and 1696.8 μmol EC/g, respectively [50]. This technology has a short extraction time, less solvent use, and higher safety, oil yield, and antioxidant activity. However, the technology has a high equipment cost and is not suitable for industrial applications. Subcritical CO_2_ combined with Soxhlet extraction overcomes the disadvantage of supercritical CO_2_ extraction by maintaining solubility under high pressure.

#### 5.1.4. Microwave-Assisted Extraction

High-frequency microwaves produce dipole eddy currents, ion conduction, and high-frequency friction; generate a large amount of heat in a short time; and accelerate solvent penetration into the material. When the microwave power was 200–560 W, the microwave time was 30–120 s, the solvent bran ratio was 1.6–3 mL/g, the overall oil quality obtained using microwave-assisted extraction was better than that obtained using conventional solvent extraction, the α-tocopherol and antioxidant activities were significantly increased, the free fatty acids were significantly decreased, and the extraction speed was faster [51]. Using ethanol, d-limonene, and other green solvents as substitutes for n-hexane, the microwave-assisted extraction of rice bran oil afforded the highest yield of 24.64% and consumed less energy [52]. Microwaves can reduce or even deactivate lipase activity in rice bran and do not destroy fatty acids, cereals, and other nutrients in rice bran oil, making it a promising alternative to the traditional method.

#### 5.1.5. Subcritical Fluid Extraction

Using the principle of phase dissolution, fat-soluble components are extracted via molecular diffusion between materials and subcritical fluids. Under the conditions of 40 °C, 20 MPa, and 120 min, the oil yield of rice bran was 17.19% [53]. The oxidative induction time and α-tocopherol content of safflower seed oil extracted using SFE were significantly increased [54]. It simultaneously inactivates lipase and extracts stable rice bran oil within a short residence time.

#### 5.1.6. CO_2_ Expansion Liquid Extraction

High-pressure CO_2_ was added to a small amount of organic solvent to promote the rapid expansion of its volume and extraction of lipids. Under the conditions of a temperature of 25 °C, pressure of 5.1 MPa, CO_2_ molar fraction of 0.87, and CO_2_ expansion of hexane 0.2 mol, the oil yield (25%) of rice bran was significantly higher than that of n-hexane extraction (20%), the phosphorus content was 50 times lower than that of n-hexane extraction, and the FFA concentration was reduced by 17% [55]. At 25 °C, 5.0 MPa, and a CO_2_ molar fraction of 0.76, a phosphorus concentration of 4.2 ppm, oil yield of 23.7%, and FFA of 9.60 wt% were obtained [56]. This extraction method has good gas solubility, mass transfer, low viscosity, low condition temperature, and a smaller amount of organic solvent. Thus, rice bran oil can be extracted without impurities or refining.

### 5.2. Refining Rice Bran Oil and Extraction of Bioactive Substances

The high acid value and dark color of rice bran crude oil make it difficult to refine rice oil. Nutrients are readily lost during the refining process. In addition, the production of trans-fatty acids, 3-chloropropanol ester, and other potentially harmful substances affects the quality and safety of rice oil products. The rice oil refining process generally involves deacidification, decolorization, deodorization, and triglyceride removal (Table 4). The byproducts of rice bran oil refinement can also be used to extract bioactive substances, such as γ-oryzanol and tocopherol.

#### 5.2.1. Deacidification

Rice bran oil usually contains many FFAs. Their removal has always been a challenge in the refining process. Alkali refining and water steam distillation deacidification are widely used in industry, but glycerolysis side reactions can easily occur. In one study, highly acidic rice bran oil was deacidified with a glyceryl eutectic solvent. The deacidification rate reached 94.5%, and the glycerolysis side reaction was prevented [57]. Magnetic carrier starch nanoparticles (MDSNP) were obtained by combining dialdehyde starch nanoparticles (DSNP) with modified Fe_3_O_4_. Magnetically immobilized Candida antarctic lipase B (MDSNPCALB) was obtained by cross-linking the enzyme with the carrier. MDSNPCALB was added to the degummed rice bran oil. Ethanolamine was used as an acyl receptor in the acylation and deacidification reactions. After ten repeated uses, MDSNPCALB maintained high activity and could be effectively used to deacidify rice bran oil [58]. When CALB was fixed on hydrophobic ordered mesoporous silicon, its catalytic activity increased by 6.6 times, temperature tolerance increased by 20 °C, 91.4% FFAs were removed in a series continuous flow enzyme reactor, and plant sterol esters and diglycerides were increased by 9 and 12 times. The reported retention rate of γ-oryzanol was at least 40% higher than that of traditional alkali refining [59]. Polyedopamine/hydroxyethyl methacrylate/acrylamide hydrogel microspheres were deacidified through esterification with a fixed lipase as the catalyst. A deacidification rate of up to 98.19% was achieved in 2 h, which was economical and sustainable [60].

#### 5.2.2. Decolorization

In one study, the decolorization rate of rice bran oil was 97.08%; the retention rates were 89.62% for glutamic acid, 90.16% for sterol, and 79.91% for vitamin E; and the adsorption rate of benzo (a) pyrene was 95.98% with a 5% compound decolorization agent (active clay/activated carbon = 5:1) [61]. Rice bran oil is rich in nutrients that benefit the human body. However, the loss of nutrients is serious because of the difficulty of bleaching. The color and quality of rice bran oil were reportedly significantly improved using ultrasonic-assisted adsorbents (floroli silica soil, magnesium trisilicate, and active blended soil), whereas the composition of micronutrients and fatty acids was maintained within an acceptable range [62].

#### 5.2.3. Deodorization

Rice bran oil contains a large amount of phytosterols, which are naturally active substances with various active functions that are beneficial to the human body. Deodorization during refining affects the content of phytosterols. A comparison of the phytosterol content in rice bran oil after deodorization with nitrogen and water vapor showed that nitrogen as a stripping gas promoted the formation of phytosterol esters, reduced the production of phytosterol oxidation products, and was more suitable for deodorizing rice bran oil [63]. The addition of sesamol to rice bran oil during deodorization inhibited glycerol formation. When 0.05% sesamol was added, the inhibition rate of glycerol was close to that of 0.02% tert-butylhydroquinone [64].

#### 5.2.4. Glyceride Removal

Rice furoic acid oil is a byproduct of rice bran oil extraction. It comprises FFAs, triglycerides, and a large amount of the natural antioxidant γ-oryzanol. Before the recovery of γ-glutamate, enzymatic hydrolysis is an alternative to the traditional base-catalyzed green triglyceride removal method with a removal rate of 99%. Subcritical water hydrolysis at 220 °C for 10 min can achieve a high triglyceride removal rate of >95%, and the content of γ-oryzanol is still acceptable (50%) [65].

#### 5.2.5. Extraction of γ-Oryzanol and Tocopherol

γ-Oryzanol and tocopherol can be extracted from the fatty acid distillation residue of the rice bran oil refining process. The concentration of tocopherol was reportedly increased by approximately six times, with a recovery rate of 51.50% using methanol and ethyl ketone (1:1). The recovery rates of tocopherol and γ-oryzanol were 92.15% and 84.12%, respectively, and the initial concentrations of these phytochemicals were increased by approximately three times [66]. γ-Tocotrienol and delta-tocotrienol are the most effective natural radiation protection agents, which can be extracted from the distillate of rice bran oil deodorant with 100% purity through the gradient elution of hexane-acetic acid (99.1:0.9) and ethyl acetoacetic acid (99.1:0.9) [67]. The byproduct of the deodorized distillate from rice bran oil contains squalene (approximately 8%) and phytosterol (approximately 4%) as unsaponifiable substances. Phytosterol and squalene can be obtained with 97% purity through supercritical fluid extraction combined with a solvent [68]. 

## 6. Application of Rice Bran Oil in Food

Rice bran oil has high stability, nutritional value, and gel properties, Therefore, it is widely used in food(Table 5). The structure, concentration, and synergism of γ-glutenin, polyphenols, phospholipids, phytosterols, and squalene give rice bran oil its unique properties [69].

### 6.1. Improved Quality of Other Edible Oils

The addition of 25% to 75% rice bran oil to soybean oil can inhibit oxidation and degradation. FFA and PV remain within the recommended range for human consumption during 12 months of storage, and thiobarbituric acid shows a similar trend to PV [70]. The thermal stability of rice bran oil and soybean oil in deep-oil frying of French fries has been studied. In the first four heating cycles, the peroxide value of the two oils remained below 10 meq/kg, and the acid value (<0.6 mg KOH/100 g) in the fifth cycle of rice bran oil and the third cycle of soybean oil met the requirements. FFA and total polar material showed significant stability after repeated heating, indicating the recoverability of the two oils. The total saturated fatty acid content increased in rice bran oil, the unsaturated fatty acid content decreased, and the stability was enhanced, indicating that rice bran oil was more stable during the frying process [71]. When rice bran oil was added to flaxseed oil at concentrations of 500, 1000, and 1500 ppm, the antioxidant capacity of flaxseed oil also increased with the increase in rice bran oil [72].

### 6.2. Preparation of Gel to Replace Fat

With the development of gel technology, liquid vegetable oils have been converted into solid fats, which have the functional and textural properties of solid fats and can be used as potential substitutes for saturated and trans-fats. The mixture of β-sitosterol and γ-oryzanol in rice bran oil can be used as a gel to construct an oil gel, which self-assembles to form a fibrous network structure in the oil phase mainly through hydrogen bonding and van der Waals forces (Figure 5). In the case of sterols only, the crystal structure of the oil gel sample consists mainly of large flakes, and excessive sterols interfere with the self-assembly of the three-dimensional gel network structure (Figure 5a). With the addition of γ-oryzanol, the liquid oil is fixed by the spherular-shaped aggregate structure formed by the two, and the network structure is composed of sterol flake crystals (Figure 5b). As the proportion of γ-glutamate is increased, at a certain proportion, the two can form one-dimensional tubules through self-assembly. These crystals grow radially to form a three-dimensional network space dominated by a spherulite shape with a dense structure, and liquid oil is deposited in an orderly manner in the lattice to form a solid oil gel (Figure 5c).

Oil gels have been studied and applied to many types of foods. In a study on the processing of cookies with soy wax/rice bran oil gel instead of butter, the dough became firmer as the amount of added oil gel increased. The prepared cookies had brown edges and light-colored centers. In addition, with an increase in oil gel content, the surface cracks of cookies increased correspondingly. Replacing up to 50% of butter with oil gel, which was feasible, had no significant effect on the physicochemical properties of cookie dough and cookies [73]. Replacing animal fat with oil gels composed of 25%, 50%, and 75% rice bran wax and rice bran oil can improve the textural, sensory, and nutritional properties of Thai sweet sausages. The higher the replacement ratio, the lower the cholesterol level (*p* < 0.05). There was no significant difference in total unsaturated fatty acid levels between the 50% and 75% oil gels. The optimal replacement ratio of oil gel was determined to be 50% [74]. The new oil gel was developed using sesame oil, rice bran oil, beeswax, and stearic acid and had similar properties to margarine. The binding rate of oil was 99.99%, showing higher oxidation stability than margarine, and a 3:1 ratio of beeswax and stearic acid synergistically affected the properties of the oil gel [75]. By adding 50% rice bran oil instead of shortening, the physical properties and sensory acceptability of the bread maximally increased the quality of the bread [76]. The PV of oil gel prepared from mixed oil, such as rice bran oil and rice bran wax, increased only slightly after 60 days of storage at 6 to 7 °C. Therefore, mixed oil-based gels can be used as new structural oil substitutes for chocolate sauce [77].

### 6.3. Supplementary Nutrition

As one of the main byproducts of the rice bran oil refining industry, rice bran gum is mainly composed of phospholipids, which can be made into powdered rice bran lecithin after bleaching with sodium chlorite to relieve the marketplace pressure of phospholipid demand [78]. Rice bran oil and mango kernel fat were used to make a cocoa butter substitute, with a 20% improved overall sensory acceptance and taste similar to that of control chocolate [79]. A double emulsion prepared from rice starch, rice protein isolate, and rice bran oil was reportedly used as a carrier, filler, and binder to enhance the physical, functional, and sensory properties of food [80]. The viscosity, permeability, and water-holding capacity of yogurt supplemented with rice bran oil and soybean protein nanoparticles decreased with decreasing size of the nanoparticles. Compared with control yogurt, the yogurt supplemented with soybean protein nanoparticles showed stronger anti-free radical scavenging ability and reduced iron antioxidant properties [81]. Rice bran oil, curcumin, and medium-chain triglycerides were ultrasonically treated into rice bran oil bodies, which can effectively transfer hydrophobic nutrients. The 1.5 wt% preparation had the highest encapsulation rate (87.67%), particle size (190 nm), storage stability, and bioavailability (61.04%) [82].

## 7. Physiological Functions of Rice Bran Oil

Rice bran oil is rich in tocopherol, γ-glutamic acid, phytosterol, and unsaponifiable substances, which are essential in antioxidant, anti-inflammatory, cancer prevention, and nerve protection functions. These constituents improve the nutritional quality and promote the value-added utilization of rice bran oil (Figure 6).

### 7.1. Antioxidant and Anti-Inflammatory Activities

The antioxidant and anti-aging ability of physically refined rice bran oil was reported to be significantly higher than that of dissolved γ-oryzanol, α-tocopherol, and sitosterol, and was superior to the mixture, indicating that the antioxidant and anti-aging effects of physically refined rice bran oil were superior to those of the antioxidants derived from rice bran oil [83]. The extraction of rice bran oil by hot pretreatment and cold pressing can improve oil yield, oxidation stability, bioactive compounds, and antioxidant activity, with a stronger inhibitory effect on cell oxidative stress induced by hydrogen peroxide [84].

Using purified rice bran oil as a lipid matrix model, different concentrations of α-tocopherol, γ-oryzanol, and phytosterol significantly affected oxidation stability and free radical scavenging ability. The inhibitory effect of phytosterol on α-tocopherol and hydrogen bond formation between γ-oryzanol and phytosterol was suggested to be mainly due to the influence on the degree of synergism or antagonism of interaction [85].

Rice bran oil is rich in tocotrienol. The scavenging of tocotrienol on 2,2-diphenyl-1-picrylhydrazyl free radicals was 53% at 100 mg/mL. Tocotrienol can effectively inhibit the production of reactive oxygen species, inhibit the migration of neutrophils to inflammatory sites, and upregulate the expression of the antioxidant genes superoxide dismutase and glutathione peroxidase. Inhibiting the expression of pro-inflammatory factors, tumor necrosis factor-alpha (TNF-α), interleukin (IL)-8, and cupric sulfate-induced inflammation have significant antioxidant and anti-inflammatory effects [86]. Rice bran oil is rich in polyphenols. Supplementation with 0.02% rice bran oil in lipopolysaccharide (LPS)-contaminated feed can improve the antioxidant capacity of piglets exposed to LPS, increase the concentrations of catalase and superoxide dismutase, increase the total antioxidant capacity, and decrease the plasma concentrations of diamine oxidase and malondialdehyde [87]. In another study, when elderly individuals with prehypertension were supplemented with 1000 mg of rice bran oil daily for eight weeks, the plasma levels of malondialdehyde, glutathione disulfide, and tumor necrosis factor-alpha (TNF-α) were significantly reduced, and the ratio of reduced glutathione to glutathione disulfide was significantly increased (all *p* < 0.05). The results indicate that the long-term consumption of rice bran oil can relieve oxidative stress and inflammation in older adults with prehypertension [88]. 

In vitro, rice bran oil treatment was demonstrated to significantly reduce the levels of the pro-inflammatory cytokines (IL-6 and TNF-α) in the culture supernatant of mouse cells, upregulate the secretion of the anti-inflammatory cytokine IL-10, and inhibit the transcription and activation markers of genes encoding the cyclooxygenase-2 and inducible nitric oxide synthase inflammatory mediators in CD80 and INOS cells. The expression of CD86 and MHC-II and the increase in mitochondrial respiration demonstrated that rice bran oil can regulate the inflammatory response of mouse macrophages by upregulating mitochondrial respiration [89]. Rice bran oil in male Wister rats and female Sprague–Dawley rats also showed effective anti-inflammatory, analgesic, and antiarthritic activities, demonstrating the therapeutic value of various bioactive ingredients, including γ-oryzanol [90].

### 7.2. Reduction of Blood Lipids and Prevention of Cardiovascular Diseases

Hyperlipidemia can lead to high blood pressure, high blood sugar levels, cardiovascular disease, liver damage, and atherosclerosis. The main components of rice bran oil (unsaturated fatty acids, triterpene alcohols, phytosterols, tocotrienols, and tocopherols) can lower cholesterol levels. The consumption of rice bran oil can significantly reduce serum total cholesterol, low-density lipoprotein cholesterol, and triglyceride levels [91]. After consumption, rice bran oil mainly enhances the expression of the low-density lipoprotein receptor cytochrome P450 family 7 subfamily member 1 and sterol regulatory element-binding protein 2, promotes the excretion of fecal cholesterol and bile acid, and decreases β-hydroxy β-methylglutaryl-CoA reductase and fatty acid synthase, thereby reducing cholesterol absorption [92]. Tocotrienol from rice bran oil extract can significantly upregulate pparγ and cyp7a1 mRNA levels, significantly down-regulate cpt1a mRNA levels (*p* < 0.01), increase the protein expression levels of Pparγ and Rxrα by two times, and reduce the synthesis and storage of adipose tissue of zebrafish [93]. The active polyphenols in rice bran oil can prevent cardiovascular diseases by improving the oxidative stress mediated by free radicals [94]. Rice bran oil extract inhibits lipid uptake in vivo. It reduces lipid accumulation in HepG2 cells, leading to the activation of AMPK and inhibition of the signal transducer and activator of transcription 3, thereby inhibiting hyperlipidemia and hepatic steatosis caused by a high-fat diet [95].

### 7.3. Cancer Prevention

After injecting diethylnitrosamine and 1,2-dimethylhydrazine into rats, rice bran oil equivalent to 100 mg/kg body weight of γ-oryzanol 5 days per week for 10 weeks inhibited precancerous lesions, including hepatic glutathione S-transferase placental positive lesions and abnormal rectum lesions. Inducing apoptosis of hepatocytes and colorectal cells and reducing the expression of pro-inflammatory cytokines also reportedly promotes diethylnitrosamine- and 1,2-dimethylhydrazine-induced changes in rat gut microbiota, such as an increase in the Firmicutes/Bacteroidetes ratio [96]. After the occurrence of colon cancer induced by 1,2-dimethylhydrazine and sodium glucan sulfate, superoxide dismutase activity and expression of the pro-inflammatory protein cyclooxygenase-2 in the colon of rats were increased when 0.8% γ-oryzanol and unsaponifiable matter extracted from rice bran oil were added to the diet. Chemically induced colon cancer in rats may be inhibited by antioxidant and anti-inflammatory mechanisms, and unsaponifiable compounds may be more effective than gamma-glutamate [97].

### 7.4. Neuroprotective Effect

1,3-Dipalmitoyl-2-oleyl glycerol (POP) is a triacylglycerol ester present in rice bran oil. Oral administration of 1 5 mg/kg POP prevents middle cerebral artery occlusion/reperfusion-induced glutathione depletion and lipid oxidative degradation in rat brains. It also significantly inhibited the phosphorylation of p38 in the ischemic brain up-regulation of MAPKs, inflammatory factors (inducible nitric oxide synthase (i-NOS) and cyclooxygenase-2 (COX-2)), and pro-apoptotic proteins (B-cell lymphoma-2 (Bcl-2) associated X protein (Bax) and lysed caspase 3) and the down-regulation of anti-apoptotic protein (Bcl-1). The down-regulation of phosphatidylinositol 3′-kinase (PI3K), phosphorylated protein kinase B (Akt), and phosphorylated ring (adenosine monophosphate) AMP response element-binding protein (CREB) in an ischemic brain was inhibited, suggesting that POP may play a neuroprotective role by inhibiting p38 MAPK and activating the PI3K/Akt/CREB pathway [69]. Phytosterols are important bioactive compounds in rice bran and rice bran oil and are found in rotenone-treated Caenorhabditis elegans. Sitosterol, ferulic acid, 24-methylene-ferulic acid, and rapeseed ester in rice bran oil activate the DAF-16/FOXO pathway and increase the neuroprotective effect of oxidative stress resistance induced by the overexpression of the cell death protein 3 apoptosis protein [98].

## 8. Challenges, Limitations, and Future Research Directions

Rice bran, which is a byproduct of rice processing, has a large annual output. However, because of its low cost and high susceptibility to spoilage (3–5 days), it is mainly used to feed animals, with only 2–5% used as a raw material for rice bran oil processing. With increasing economic development and health awareness, individuals are increasingly paying attention to the health benefits and nutritional properties of vegetable oil, which is recommended for its rich nutritional content. Different types of vegetable oils can be selected according to individual physical conditions and nutritional needs to supplement more unsaturated fatty acids and plant sterols. In terms of the nutritional quality evaluation of rice bran oil, the analysis of bioactive components has been limited mainly to the analysis of γ-oryzanol, phytosterols, and lipids. Analyses and the identification of hitherto unknown small molecule active substances are lacking. Therefore, the health benefits of rice bran oil require further investigation. Rice bran oil is rich in unsaturated fatty acids and various bioactive components, which have been confirmed by many researchers. However, because it has a higher price than soybean oil, a unique odor, and insufficient consumer awareness, its consumption is limited. Therefore, research on low-cost and high-quality rice bran oil processing technologies and supporting equipment, as well as consumer awareness promotion and education, are effective ways to increase rice bran oil consumption.

N-hexane solvent extraction has become a mainstream production technology in the rice bran processing industry due to its low cost. However, from the perspective of nutritional characteristics and safety, under the premise of ensuring the yield and quality of rice bran oil, increased research of emerging extraction and auxiliary technologies is needed, as is an exploration of the potential combined use of the technologies. With technological developments, new rice bran oil extraction technologies have emerged. The efficiency and superiority of the extraction were mainly investigated in the experimental research stage. Owing to the immaturity of the production equipment and higher equipment costs, it is difficult to apply new technologies in production. Therefore, future research should focus on efficient and low-cost emerging technologies that support equipment production. Ultrasound-assisted extraction is very promising and highly recommended. This technology is characterized by low temperatures and fast extraction, does not change the chemical composition and structure of nutrients, and more importantly, promotes the extraction of active substances. 

With the gradual penetration of Western eating habits, foods containing light cream are becoming increasingly preferred by Chinese people. In 2022, the output of China’s margarine industry was approximately 192.75 million tons, with a demand of approximately 2.5297 million tons. Commercial butter is primarily plant-based cream, accounting for approximately 80% of the total. Traditional vegetable cream (hydrogenated oil) has a high content of trans-fatty acids, which increases the risk of cardiovascular disease, diabetes, and other diseases. The use of trans-fatty acids has been restricted in many countries worldwide. The mixture of β-sitosterol and γ-oryzanol in rice bran oil can be used as a gel to construct an oil gel, which can convert liquid vegetable oil into solid fat and can be used as an ideal alternative to traditional vegetable cream and improve the nutritional properties of the oil gel. Other applications of rice bran oil in the food sector have yet to be further explored.

Rice bran oil is rich in tocopherol, γ-glutamate, phytosterol, and other unsaponifiable substances, which have remarkable antioxidant, anti-inflammatory, cancer prevention, and nerve protective effects. Future studies should prioritize reducing blood sugar and lipid levels while also enhancing the intestinal environment. Correlation analyses of the mechanism of action are needed. This study provides a scientific basis for the functional evaluation and application of rice bran oil.

## Figures and Tables

**Figure 1 foods-13-03262-f001:**
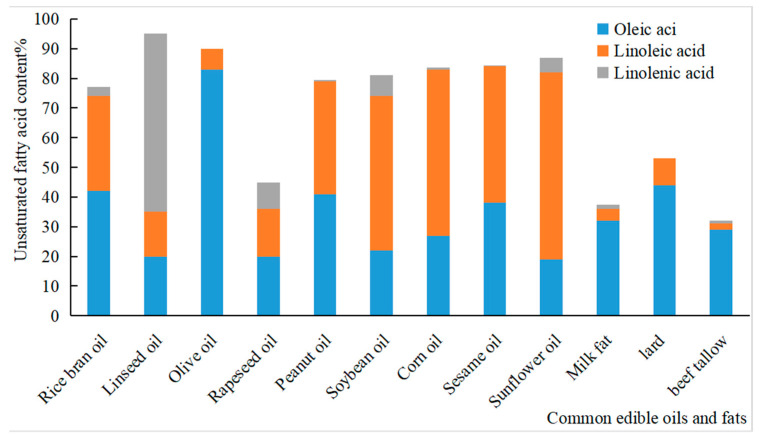
Contents of unsaturated fatty acids in common edible fats (average value).

**Figure 2 foods-13-03262-f002:**
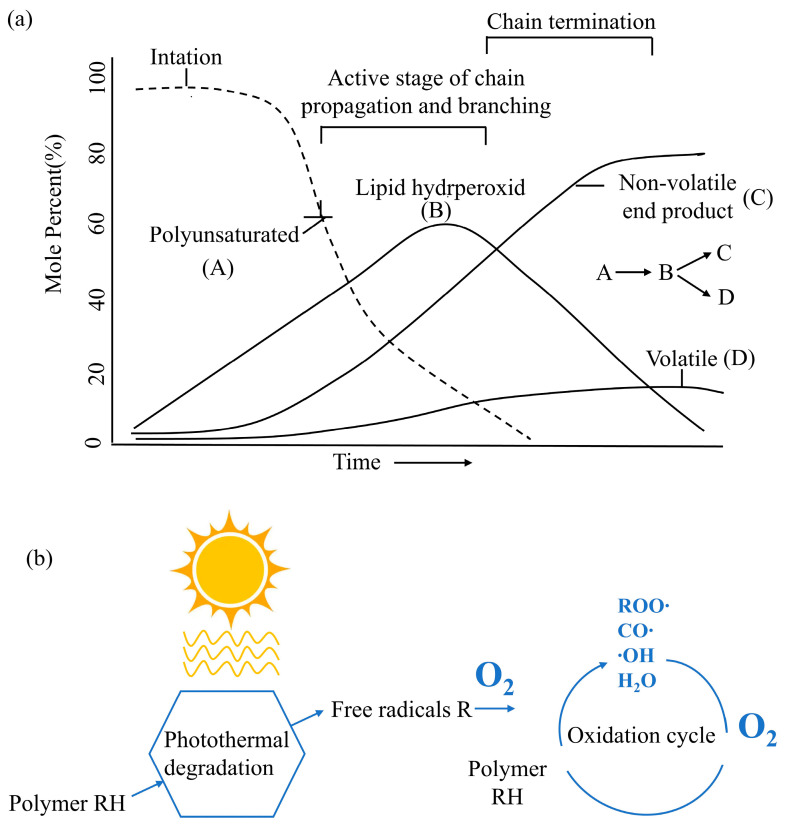
Three mechanisms for oil oxidation: (**a**) automatic oxidation; (**b**) photothermal oxidation; and (**c**) enzymatic hydrolysis and β-oxidation.

**Figure 3 foods-13-03262-f003:**
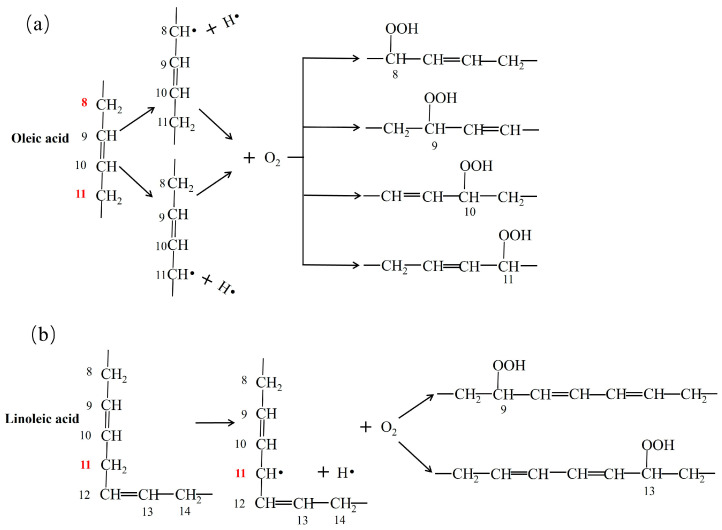
Oxidation mechanism for oil: (**a**) oleic acid; (**b**) linoleic acid.

**Figure 4 foods-13-03262-f004:**
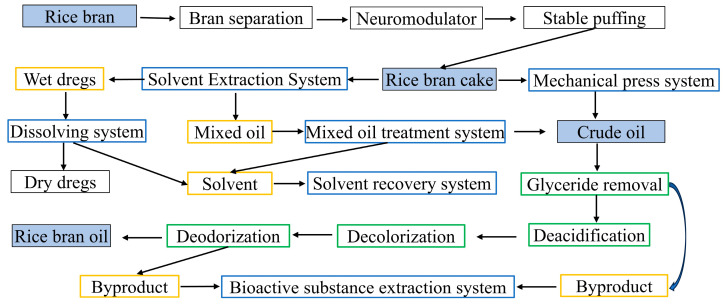
Two extraction processes for rice bran oil.

**Figure 5 foods-13-03262-f005:**
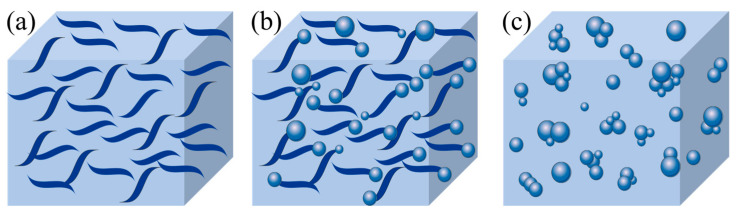
Preparation principle of sterol- and glutamate-mixed oil gels: (**a**) sterols only; (**b**) with the addition of γ-oryzanol; and (**c**) from a solid oil gel.

**Figure 6 foods-13-03262-f006:**
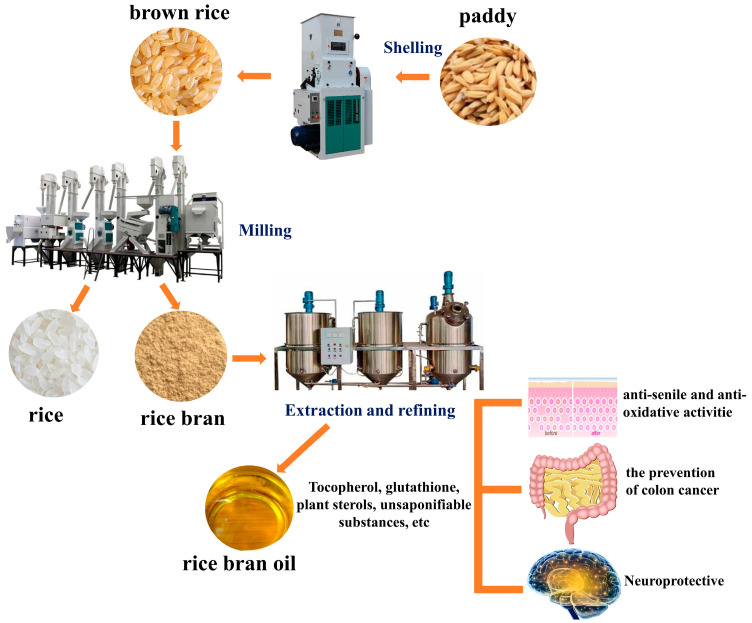
Extraction process and physiological function of rice bran oil.

**Table 1 foods-13-03262-t001:** Main nutrients of rice bran oil.

Nutrient Composition	Content	Reference
Triglyceride	81~84%	[9]
Diglycerol	2~3%	
Monoglycerin	1~2%	
Free fatty acid	2~6%	
Wax	3~4%	
Glycolipid	0.8%	
Phospholipid	1~2%	
γ-Glutamin	0.9~2.9%	[10]
Tocopherol	0.10~0.14%	
Phytosterol	2~3%	
Sterol ester	2.8~3.1%	
Saturated fatty acid	23.63~24.7%	[11]
Palmitic acid	13.9~27.4%	
Stearic acid	1.5~4.7%	
Monounsaturated fatty acids	43.71~44.3%	
Oleic acid	35.9~49.2%	
Polyunsaturated fatty acids	31~33.6%	
Linoleic acid	27.3~41.0%	

**Table 2 foods-13-03262-t002:** Techniques and characteristics of rice bran stabilization.

Stabilization Technique	Advantages and Disadvantages
Vapor treatment	The stabilization effect is general but increases the moisture content and requires further drying.
Hot air drying	It reduces the moisture content of rice bran, and the stabilization effect is better, but the treatment time is longer.
Microwave treatment	The stabilization effect is better, and the content of active ingredients can be increased, but the uniformity of treatment needs to be further improved.
Infrared radiation	The lipase inhibition rate is very high, and the stability effect is very good, but the uniformity needs to be improved.
Coupled processing	The stabilization effect is the best, but the processing cost is higher.

**Table 3 foods-13-03262-t003:** New rice bran oil extraction technology.

Extraction Technology	Advantages and Disadvantages
Enzyme-assisted extraction	The reaction conditions are relatively mild, do not damage the quality of rice bran oil, and can also improve the availability of physiologically active components. However, enzyme activity and stability need to be further improved.
Ultrasonic-assisted extraction	It has a low temperature, fast extraction rate, and no change in the chemical composition and structure of nutrients and can also promote the extraction of active substances.
Supercritical and subcritical CO_2_ extraction	The extraction time is short, the solvent usage is low, and the oil yield and antioxidant activity are higher, but the equipment cost of this technology is relatively high.
Microwave-assisted extraction	Lipase activity inhibition is effective and does not damage nutritional components, making it a promising method for extracting rice bran oil, but be careful to control the microwave power and temperature.
Subcritical fluid extraction	It can simultaneously inactivate lipase and extract rice bran oil in a short period of time, but equipment costs are high.
CO_2_ expansion liquid extraction	Mild conditions and the minimal use of organic solvents allow for the extraction of impurity-free rice bran oil without the need for refining, but equipment costs are high.

**Table 4 foods-13-03262-t004:** Refinement and extraction of active substances.

Processing	Functions
Deacidification	Remove free fatty acids
Decoloration	Adsorption decolorization, bleaching
Deodorization	Nitrogen removes odors
Glyceride removal	Remove triglycerides
Extraction of γ-oryzanol and tocopherol	Extracting active ingredients from refined byproducts

**Table 5 foods-13-03262-t005:** Application of rice bran oil in food.

Application	Advantages and Disadvantages
Add to other oils	Enhance stability
Fat replacement	Preparation of gel instead of saturated fat
Nutritional supplementation	Supplement plant sterols, etc.

## Data Availability

No new data were created or analyzed in this study. Data sharing is not applicable to this article.

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
