# Peer review of "Research Progress on the Quality, Extraction Technology, Food Application, and Physiological Function of Rice Bran Oil"

_foods, 2024, doi:10.3390/foods13203262_

Round 1

Reviewer 1 Report

Comments and Suggestions for Authors

 The work is a review and its scope is appropiate. Fig 1 the authorsgive butter, it should be milk fat and this please be corrected. Butter is an emulsion and contain water and milk fat. Fatty acids refer to milk fat. Is the data on fatty acids are averages, please specify this to be clear.                                                                                                                                                                                                    

Comments on the Quality of English Language

English languge is good

Author Response

Comment 1:The work is a review and its scope is appropiate. Fig 1 the authorsgive butter, it should be milk fat and this please be corrected. Butter is an emulsion and contain water and milk fat. Fatty acids refer to milk fat. Is the data on fatty acids are averages, please specify this to be clear.    

Response 1: Thank you for pointing this out. I agree with this comment. Therefore, I have changed 'butter' to 'milk fat' in Figure 1 and added 'average' in the caption.

Reviewer 2 Report

Comments and Suggestions for Authors

The current review article entitled "Research progress on comprehensive quality, extraction technology, food application, and physiological function of rice bran oil" investigated the nutritional and safety qualities of rice bran oil, as well as the stabilization of rice bran raw materials, extraction and refining process of rice bran oil, and discussed food applications and sub-health regulations. Despite being interesting in topic, however, it needs major improvement before further consideration:

1. The methodology of the study is not clear. Please add a heading to describe your criteria for selecting studies and how many you used in your review.

2. Each heading should have its detailed Table/Figure. For an instance, in 4. Extraction technology of rice bran oil, you should summurize each extraction in Table and describe the limitations and benefits. Please apply this to all headings: 3,4,5,6,......

3. Highly suggest to add a heading to describe Challenges and Limitations as well as Future Research Directions before the conclusion section. 

My specific comments are given in a pdf file. 

Thank you.

Comments on the Quality of English Language

Moderate improvement seems necessary. 

Reviewer 3 Report

Comments and Suggestions for Authors

Research progress on comprehensive quality, extraction tech-2 nology, food application, and physiological function of rice 3 bran oil

 The paper is well written, but it lacks information regarding the methodology used for writing the review. More flowcharts/figures should be added. The authors addressed many topics related to rice bran oil, without going into much depth on each of them.

 Abstract

Comment 1:  Add the main fatty acids present in rice oil for human health.

Comment 2:  Lines 27-29: What was the review methodology used? Briefly describe how it was conducted.

Comment 3:  I suggest reducing the amount of information or improving the flow of the writing by adding better connectivity between the sentences.

Keywords

Comment 4: Do not use keywords that are described in the title.

 Introduction

Comment 5:

Lines: 34-35: “Rice is the world's main cereal crop, with approximately half of the world's popula-34 tion consuming it as a staple food.”

Lines: 42-45: “Rice is the world's leading cereal crop, and serves as the staple food for approximately half of the global population”

Remove one of the duplicate sentences.

 Comment 6: There are many duplicate sentences in the introduction. I suggest reviewing the writing.

 Comment 7: A subtopic titled 'Review Structure' should be created after the introduction, which should describe the databases used, search keywords, number of papers reviewed, and other relevant information.

 2. Nutritional quality of rice bran oil

 Comment 8: I suggest focusing more on the description of nutritional quality and leaving the effects of processing for another topic

 3. Safety quality of rice bran oil

 Comment 9: Lines 154: I suggest creating a flowchart/diagram detailing the three mechanisms of rice oil oxidation.

 4. Extraction technology of rice bran oil

 Comment 10: I suggest initially describing the techniques (mechanical, solvent, and mixed) before reporting studies that analyzed the extractions.

 Comment 11: It would be interesting to add a flowchart/figure of the main techniques used.

 5 Refining rice bran oil and extraction of bioactive substances

 Comment 12: Add tables/figures of the oil refining stages.

 Author Response

Please refer to the attachment for the reply.

Round 2

Reviewer 3 Report

Comments and Suggestions for Authors

After the authors' corrections, I suggest publishing the current format.

Author Response

Please see annex for details。